# Self-selection of dissipative assemblies driven by primitive chemical reaction networks

Marta Tena-Solsona[1,2], Caren Wanzke[1], Benedikt Riess[1], Andreas R. Bausch[3] & Job Boekhoven [1,2]

Life is a dissipative nonequilibrium structure that requires constant consumption of energy to sustain itself. How such an unstable state could have selected from an abiotic pool of molecules remains a mystery. Here we show that liquid phase-separation offers a mechanism for the selection of dissipative products from a library of reacting molecules. We bring a set of primitive carboxylic acids out-of-equilibrium by addition of high-energy condensing agents. The resulting anhydrides are transiently present before deactivation via hydrolysis. We find the anhydrides that phase-separate into droplets to protect themselves from hydrolysis and to be more persistent than non-assembling ones. Thus, after several starvation-refueling cycles, the library self-selects the phase-separating anhydrides. We observe that the self-selection mechanism is more effective when the library is brought out-of-equilibrium by periodic addition of batches as opposed to feeding it continuously. Our results suggest that phase-separation offers a selection mechanism for energy dissipating assemblies.

[1] Department of Chemistry, Technische Universität München, Lichtenbergstrasse 4, Garching 85748, Germany. [2] Institute for Advanced Study, Technische Universität München, Lichtenbergstrasse 2a, Garching 85748, Germany. [3] Lehrstuhl für Biophysik E27, Physik-Department, Technische Universität München, James-Franck-Straße 1, Garching 85748, Germany. Correspondence and requests for materials should be addressed to J.B. (email: job.boekhoven@tum.de)

In a dynamic combinatorial libraries, molecules react reversibly with one another to give a pool of products from simple precursors[1]. Driven by thermodynamic control, products are selected on the basis of their position in the free energy landscape[2]. The free energy gained upon self-assembly can thus favor an assembling product over non-assembling competitors[3–5]. The selection of a stable assembling species out of pool of non-assembling precursors is an attractive mechanism to the formation of protocells out of a pool of prebiotic molecules[6]. Indeed, selection mechanisms are also of central importance for the proper function of biological systems[7,8]. However, life does not operate in equilibrium with its environment, but is an intrinsically unstable energy dissipating state endowed with persistence[9,10]. Mechanisms should thus be sought that can select nonequilibrium species[11–13]. Recent work has demonstrated the emergence of life-like features in dissipative nonbiological systems, including bistability and oscillations[14], the formation of molecular assemblies[5,15–22] that can undergo dynamic instabilities[23,24] or even self-division[25], yet efficient selection mechanisms need to be identified.

Here we show that phase-separation is an efficient way for a controlled selection of chemical nonequilibrium species. We bring a library of primitive carboxylic acids out-of-equilibrium by high-energy condensing agents. We find the metastable anhydride products that can phase-separate into micron-sized droplets to be more persistent against deactivation than non-assembling ones. After several starvation-refueling cycles, the library self-selects the most competitive product by their survival in phase-separated droplets. The observed self-selection is rationalized by a first-order kinetic model, taking into account the persistence of species by compartmentalization. We found that it is more favorable for the self-selecting droplets to form when the library is brought out-of-equilibrium by periodic addition of batches as opposed to feeding it continuously. Our results suggest that persistence by phase-separation or compartmentalization offers a selection mechanism for energy dissipating out-of-equilibrium assemblies.

## Results

### A primitive chemical reaction network that forms nonequilibrium anhydrides.
Figure 1a depicts the chemical reaction network. We used aqueous solutions of 300 mM propionic, 300 mM butyric, 300 mM valeric, and 100 mM caproic acid as precursors ($C_3$, $C_4$, $C_5$, or $C_6$, respectively). Condensing agents 1-ethyl-3-(3-dimethylaminopropyl)carbodiimide (EDC), $N,N'$-diisopropylcarbodiimide, and 1,1′-carbonyldiimidazole (CDI) were used as fuels that drive the chemical network out-of-equilibrium. Indeed, when we added a batch of these fuels to aqueous $C_5$, the corresponding symmetric anhydride ($C_5C_5$) was temporarily found by high-performance liquid chromatography (HPLC) at the expense of the fuel (Fig. 1b, Supplementary Fig. 2 and Supplementary Table 1). We focused the further experiments on EDC, the most active condensing agent. The side-product N-acyl urea was found at low concentrations but did interfere with the networks (Supplementary Fig. 3i-l and Supplementary Table 2).

We found a drastically different evolution of the corresponding anhydride concentration for the four precursors, despite their molecular similarity. Independent on the precursor's carbon number, the time course of production of the anhydride was comparable in the presence of EDC (10 mM) with a maximum anhydride concentration of roughly 3–5 mM. After the consumption of the fuel, we observed that the decay of the anhydrides was dependent on the carbon number. The higher carbon number anhydrides hydrolyzed slower with a drastic increase of stability for the $C_6C_6$ (Fig. 1b). The dependence of the

hydrolysis on the carbon number resulted in a significant difference in the remaining anhydride after 1 h: while for $C_3C_3$ and $C_4C_4$ almost no anhydride was detected, 1.5 and close to 3.5 mM remained for the $C_5C_5$ and $C_6C_6$, respectively (Fig. 1b).

### Phase-separation inhibits anhydride hydrolysis.
The different remaining concentrations after 1 h resulted from differences in the decay profile of the anhydride, which changed from an exponential decay for $C_3C_3$ and $C_4C_4$ to a linear decay for $C_5C_5$ and $C_6C_6$. The EDC consumption rate for all precursors was similar, from which we can conclude that the anhydride formation rate was independent of carbon number (Supplementary Fig. 3e-h). We found the solutions with higher anhydride persistence to become temporarily turbid, for just over 50 min in the case of $C_5C_5$ and far over 2 h for $C_6C_6$ (Supplementary Fig. 4c-d). Further inspection revealed that the turbidity arose from phase-separation of the anhydride into droplets (Fig. 1c). Confocal microscopy revealed that the droplets emerged directly after addition of fuel. Particularly for $C_5$, the droplet radius rapidly grew from roughly 0.7 μm to a maximum of 1.0 μm at 20 min (Fig. 1c). The time where microscopy found no more droplets matched well with the time the solution became transparent again. The disappearance of the micro-phase-separation is due to lowering of the concentration of anhydride below its solubility limit. The formation of phase-separated droplets was found for all anhydrides except $C_3C_3$, although for the lower carbon number anhydride ($C_4C_4$), a higher amount of EDC (at least 50 mM) was required (Supplementary Figs. 4b and 5).

We explain the increased persistence of the anhydride with higher carbon numbers by the emergence of the phase-separated oil droplets (Fig. 1d). These droplets protect the anhydride from the water, which is required for hydrolysis. Thus, the hydrolysis takes only place on the anhydride that remains in solution which is limited by its solubility ($s_{anhydride}$). Under these conditions, the hydrolysis rate can be calculated by $v_{hydrolysis} = k_{hydrolysis} \times s_{anhydride}$ where $k$ is the hydrolysis rate constant. Since the solubility and the rate constant are constants, the resulting hydrolysis rate is a constant, too. This mechanism explains the observed linear decay of the anhydride concentration after the system had consumed all fuel. Adding the mechanism of inhibition of the deactivation by phase-separation into a kinetic model, the dynamics are recovered (See Supplementary Fig. 1, Supplementary Table 3 and Methods section for description of the model). The anhydride solubility was found to decrease roughly an order of magnitude per two carbon numbers (11, 1.0, and 0.05 mM for $C_4C_4$, $C_5C_5$, and $C_6C_6$, respectively), which is in line with findings for aliphatic carboxylic acids[26] and alcohols[27]. These solubilities matched the turbidity data well, considering that when the concentration anhydride fell below the solubility limit, we observed no more increased turbidity (Supplementary Fig. 4). The model was confirmed by performing the fit for several initial fuel concentrations without any adaptation of the parameters (Supplementary Table 3 and Supplementary Fig. 3). The different solubilities and thus hydrolysis rates depending on the carbon number act as an effective persistence of the anhydride, which we numerically assess (Fig. 1e, see Methods section for calculations). In these calculations, a persistence factor of 1.0 meant no decreased hydrolysis rate compared to without droplets, whereas a factor of 100 would imply 100-fold slower hydrolysis. As the anhydride hydrolysis rate decreased roughly an order of magnitude per carbon number, the persistence factor increased roughly an order of magnitude per increase in carbon number.

### Phase-separated anhydrides are self-selected.
The increased persistence of the anhydrides by phase-separation suggests that in

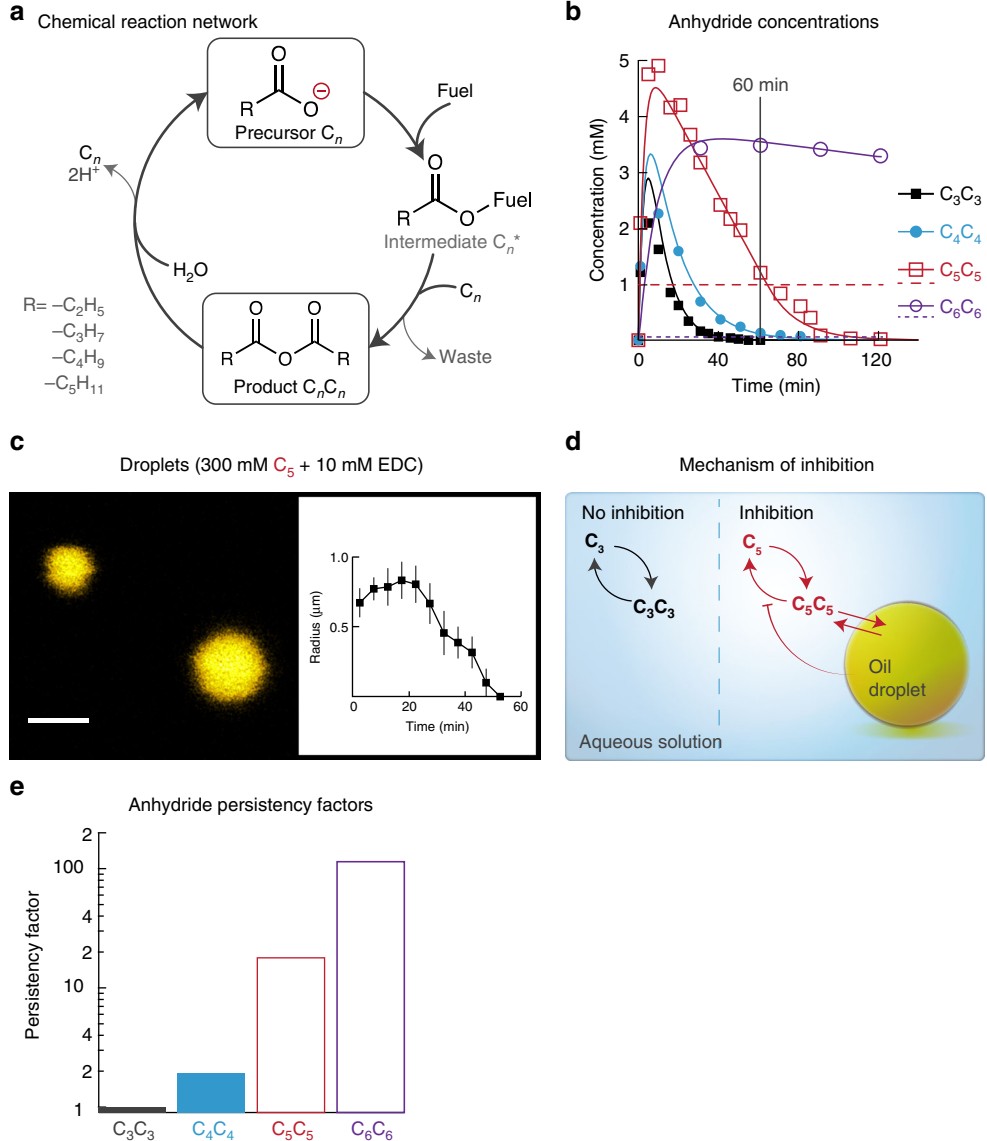

**Fig. 1** Phase-separation into anhydride-droplets inhibits anhydride hydrolysis. **a** Chemical reaction network that drives the formation of phase-separating products. Linear saturated aliphatic carboxylic acid precursors ($C_n$) are activated by high-energy condensing agents (fuel). The activated intermediate ($C_n*$) can react with a second precursor molecule to form an anhydride product ($C_nC_n$). The network operates in water and the product thus rapidly hydrolyzes to the original precursor. **b** Concentration of the corresponding anhydrides when 300 mM $C_3$ (black), 300 mM $C_4$ (blue), 300 mM $C_5$ (red), or 100 mM $C_6$ (purple) was subjected to 10 mM EDC. Markers represent HPLC data; solid lines represent data calculated by the kinetic model. Note the linear decay of the anhydride when the concentration anhydride is above its solubility. Horizontal dashed lines represent $C_5C_5$ (red) and $C_6C_6$ (purple) solubilities. **c** Confocal micrograph of 300 mM $C_5$ with 10 mM EDC after 15 min. Scale bar represents 2 μm. The inset shows the average droplet radii against time for 300 mM $C_5$ with 10 mM EDC. The solution was imaged every minute. Data of 5 min was binned for statistical analysis. Error bars refer to the standard deviation between experiments ($n = 3$). **d** Schematic representation of the mechanism of inhibition of the hydrolysis by droplets. $C_3C_3$ is well soluble and rapidly hydrolyzes. $C_5C_5$ is not well soluble and phase-separates into droplets. The phase-separated anhydride is not accessible to the $H_2O$ required for hydrolysis. Hydrolysis thus only occurs to anhydride in solution, which concentration equals the anhydride solubility. **e** Persistency factor against carbon number. The factor is calculated by the anhydride hydrolysis rate in the absence of droplets divided the anhydride hydrolysis rate in the presence of assemblies

reaction networks, higher carbon number anhydrides could be competitively selected. To test this hypothesis, we let 300 mM $C_3$ and 300 mM $C_5$ compete for batches of EDC. The chemical reaction network can form three possible products: symmetric $C_3C_3$ and $C_5C_5$, and nonsymmetric $C_3C_5$ (Fig. 2a). Addition of 5 mM EDC to the mixture resulted in the transient formation of all three anhydrides. Because $C_3C_5$ could be formed via two pathways, its concentration was highest throughout the experiment (Fig. 2b). Under these conditions, all anhydride concentrations remained below their solubility, and we thus observed no

increased turbidity. All anhydrides hydrolyzed within an hour, and the system had reset to the initial conditions. Consequently, repetitive fueling and starvation cycles resulted in the same outcome.

Next we changed the conditions to a regime, where the stabilization of products by phase-separation could occur. For example, when we added 35 mM EDC every hour, $C_3C_5$ and $C_5C_5$ both peaked at close to 7 mM (Fig. 2c). However, $C_5C_5$ persisted for much longer than its competitors, and more than 2 mM $C_5C_5$ remained when we added a new batch of fuel, while all

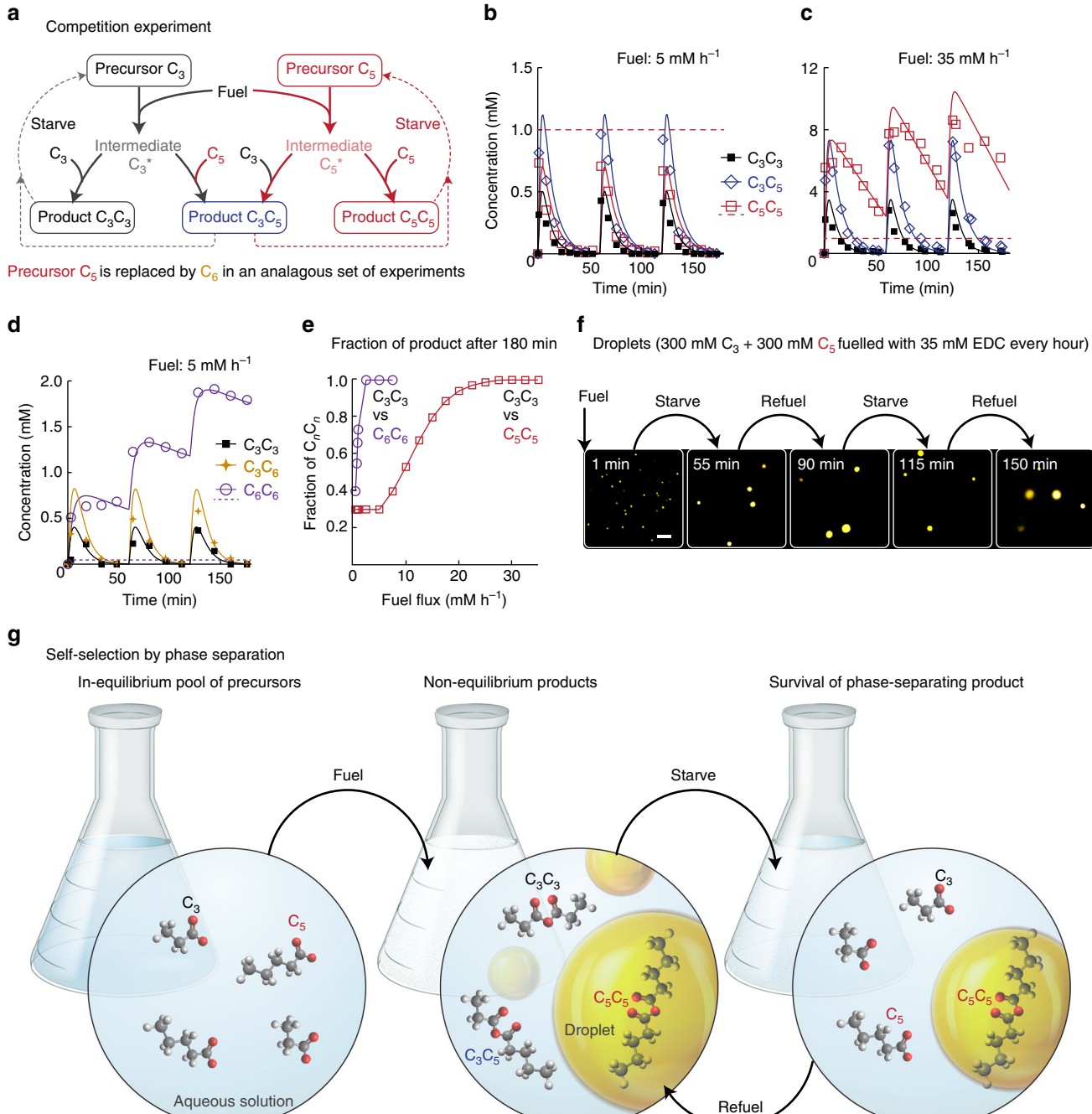

**Fig. 2** Selection of nonequilibrium phase-separating product by persistency mechanism. **a** Minimalistic representation of the chemical reaction networks in competition experiments when $C_3$ and $C_5$ are competing. $C_n$ represents carboxylic acid precursors, $C_n*$ the activated precursors and $C_nC_m$ the transient anhydride products. Three products are to be expected. Dashed arrows indicate anhydride hydrolysis reactions, whereas bold arrows depict activation reactions. The concentration of anhydride over time when 300 mM $C_3$ and 300 mM $C_5$ are fueled with 5 (**b**) or 35 (**c**) mM EDC each hour and when **d** 100 mM $C_3$ and 100 mM $C_6$ are fueled with 5 mM EDC each hour. Markers represent HPLC data; solid lines represent calculated data. Dashed lines represent the solubility of the $C_5C_5$ (red) or $C_6C_6$ (purple). **e** Fraction of the higher carbon number product after 3 h for $C_5C_5$ (red) and $C_6C_6$ (purple) against fuel flux. The red curve shows the competition between 300 mM $C_3$ and 300 mM $C_5$, whereas the purple curve shows 100 mM $C_3$ and 100 mM $C_6$. Markers represent calculated data; the connecting line is added to guide the eye. **f** Representative micrographs showing the evolution of the fueling-starvation-refueling experiment when 300 mM $C_3$ and 300 mM $C_5$ are fueled with 35 mM EDC every hour. Scale bars represent 5 μm. **g** A library of primitive carboxylic acids ($C_n$) in water is fueled by high-energy condensing agents to form a mixture of nonequilibrium anhydride products ($C_nC_m$). Products that phase-separate (e.g., $C_5C_5$) in micro-droplets were found to be most persistent and could survive starvation periods. After multiple fueling-starvation rounds, a population of self-selected persistent nonequilibrium droplets was found

competitors had hydrolyzed. Addition of a new batch of fuel further increased the advantage of $C_5C_5$, which now peaked at around 10 mM. The advantage kept on increasing with every refueling round. Confocal microscopy showed the presence of oil

droplets after addition of the first 35 mM EDC and confirmed that some droplets persisted before addition of new fuel (Fig. 2f). Refueling the experiment did not increase the number of droplets, but increased the radius of the existing droplets, as it was more

favorable for freshly formed anhydride to assemble with persisting drops, rather than forming new ones (Supplementary Fig. 7b, e). Furthermore, turbidity experiments showed that the samples remained turbid throughout the entire experiment (Supplementary Fig. 7h).

Similar competition experiments were carried out between 100 mM $C_3$ and 100 mM $C_6$, where the $C_6C_6$ has roughly two orders of greater persistence factor compared to $C_3C_3$. Because of the increased persistence factor, fueling the mixture with as little as 5 mM h$^{-1}$ was sufficient to select $C_6C_6$ (Fig. 2d). Again, a solution was obtained that remained turbid throughout the 3h experiment and contained droplets that increased in radius with each refueling round (Supplementary Fig. 7c, f, i).

Some of the $k$-values in the model were adjusted to fit the HPLC data of the competition experiments (Fig. 2b–d), which allowed us to quantify the degree of the self-selection (Supplementary Fig. 6a-c). As a measure of self-selection, we calculated the ratio of the three anhydrides after three completed fueling-starvation rounds. From the model, it followed that for competition between $C_3$ and $C_5$, at least 7 mM EDC was required to form sufficient $C_5C_5$ to peak above its solubility and thus induce the anhydride's phase-separation and increase its persistence. That implied that when the concentration of fuel is above 7 mM h$^{-1}$, the fraction $C_5C_5$ after 3 h started to increase at the expense of the fraction of $C_3C_3$ and $C_3C_5$ (Fig. 2e). Further increasing the amount of fuel added at each spike resulted in a higher advantage for $C_5C_5$. For competition between $C_3$ and $C_6$, the amount of fuel required to start increasing the advantage for the $C_6C_6$ anhydride was as little as 0.5 mM EDC every hour (Fig. 2e).

The combined results imply that the key mechanism of product selection is the protection by the product-specific formation of a droplet population, which survives the starvation periods (Fig. 2g). It is therefore only the solubility of the products that determines the persistence of a given species. At each refueling round, the concentration of the persisting anhydride increases further. These droplets increase in radius with each refueling round (Supplementary Fig. 7b, c). To demonstrate the generality of the self-selection principle, we carried out a competition experiment in which $C_3$, $C_4$, $C_5$, and $C_6$ were all precursors competing for repetitive fuel injections (5 mM h$^{-1}$ EDC). In this reaction network, we could expect ten possible anhydride products. Many of the products were isomers of one another (e.g., $C_4C_6$ and $C_5C_5$), which made resolving each product by HPLC impossible. However, we were able to quantify the concentrations of $C_3C_3$, $C_3C_4$, and $C_5C_6$, and $C_6C_6$, since they had no isomers. We quantified other products per isomer group (e.g., $C_{10}$ group: $C_4C_6$ and $C_5C_5$). Addition of 5 mM of EDC every hour to the mixture of precursors resulted in a turbid solution. The two products with the highest carbon number ($C_6C_6$ and $C_5C_6$) out of the ten possible products persisted to survive a 1h starvation period and accumulated with each refueling round. The findings demonstrated that, again, the highest carbon number anhydrides were selected (Supplementary Fig. 6d).

**Influence of the frequency of fueling in selection**. In a final set of experiments, we tested the influence of the frequency of the fuel delivery on the anhydride self-selection. Each experiment contained a mixture of 300 mM $C_3$ and 300 mM $C_5$. We added a total amount of 60 mM EDC over 3 h. Crucially, we fueled each experiment at different frequencies: either every hour 20 mM batches were added (1 h$^{-1}$), every 6 min 2 mM batches were added (10 h$^{-1}$), or 60 mM was added in continuously over the 3h experiment by a microsyringe pump ($\infty$ h$^{-1}$, Fig. 3a). We found no increased turbidity at the higher fueling frequencies, be it

continuous addition or addition of 2 mM batches every 6 min (10 h$^{-1}$). The $C_5C_5$ anhydride concentration plateaued just below its solubility (Fig. 3b and Supplementary Fig. 6e) and was not the winning anhydride with a fraction of 0.3 of the total amount of anhydride (Fig. 3d). In contrast, at a lower fueling frequency of a 20 mM batch each hour (1 h$^{-1}$), the solution turned turbid, and we found droplets (Supplementary Fig. 7a, d, g). Indeed, the concentration $C_5C_5$ peaked above its solubility and persisted for far longer than all its competitors (Fig. 3c and Supplementary Fig. 6f-h). In fact, for most of the duration of the experiment, it was the anhydride with the highest concentration.

The mechanism of the frequency-depended selection is the threshold concentration of anhydride that needs to be surpassed to enable the phase-separation of the product, which in turn increases its persistence. This frequency-dependence is most evident by varying the frequency of fuel injection, but keeping the total added amount of fuel over the entire experiment constant (Fig. 3d). When a relatively large batch of fuel is added at a low frequency (i.e., above 4 mM, five times per hour, 5 h$^{-1}$) the production of $C_5C_5$ is sufficient to pass its solubility and to phase-separate. Thus, the hydrolysis of $C_5C_5$ is decreased ensuring its concentration remains above the solubility for a prolonged time. In contrast, if we added smaller batches more frequently, a pseudo-steady state is created in which none of the anhydrides phase-separates, and thus no selection is induced.

## Discussion

In conclusion, our results show that the ability of phase-separation is an effective mechanism for the selection of none-quilibrium products, even in complex reaction schemes with tens of interacting reactions. Selection is achieved by inducing compartmentalization, which results in separating the time scales of deactivation and therefore slowing down the deactivation of products. Compartmentalization by phase-separation has already been shown to be an essential mechanism in biology for cells for proper function[28], and our results suggest that there also a more generic role of this mechanism in primitive chemical reaction networks. Moreover, work by others has shown that driven phase-separation can result in spontaneous self-division[25], as well as self-propulsion[29]. Importantly, we can extend the governing principles of the selection to other self-assembling compartmentalization mechanisms, including the formation of vesicles[30,31], polymer coacervates[32], or even simple processes such as the hybridization of RNA[33,34]. The simplicity of the inhibition mechanism will enable the further design of reaction schemes to self-select assemblies with life-like features from molecular libraries driven out-of-equilibrium.

## Methods

**Materials**. 4-Morpholineethanesulfonic acid (MES) buffer, carboxylic acids, anhydrides, fuels, and Nile Red were all purchased from Sigma-Aldrich and Alfa-Aesar unless indicated otherwise. We synthesized the different $N$-acyl ureas by repetitive additions of EDC to the precursors. We then purified the $N$-acyl ureas using reversed-phase-HPLC (Thermo Fisher Dionex Ultimate 3000, Hypersil Gold 250 × 4.8 mm) and assessed their purity with electrospray ionization–mass spectroscopy (ESI–MS) and HPLC.

**Sample preparation**. We prepared stock solutions of the precursor by dissolving the precursor in 500 mM MES buffer and adjusting the pH to 6.0. We prepared stock solutions of EDC by dissolving the EDC powder in MQ water. Typically, we used freshly prepared stock solutions of 1.0 M EDC. In addition to the high concentration EDC to the precursor solution started the cycles. We found that the networks were relatively temperature sensitive, and thus carried them out as close to 25 °C as possible.

**Batch fueled kinetic experiments**. The noncompetition experiments were carried out by addition of various batch sizes of EDC to 300 mM $C_3$, 300 mM $C_4$, 300 mM $C_5$ or 100 mM $C_6$. All competition experiments of $C_3$ and $C_5$ were carried out with

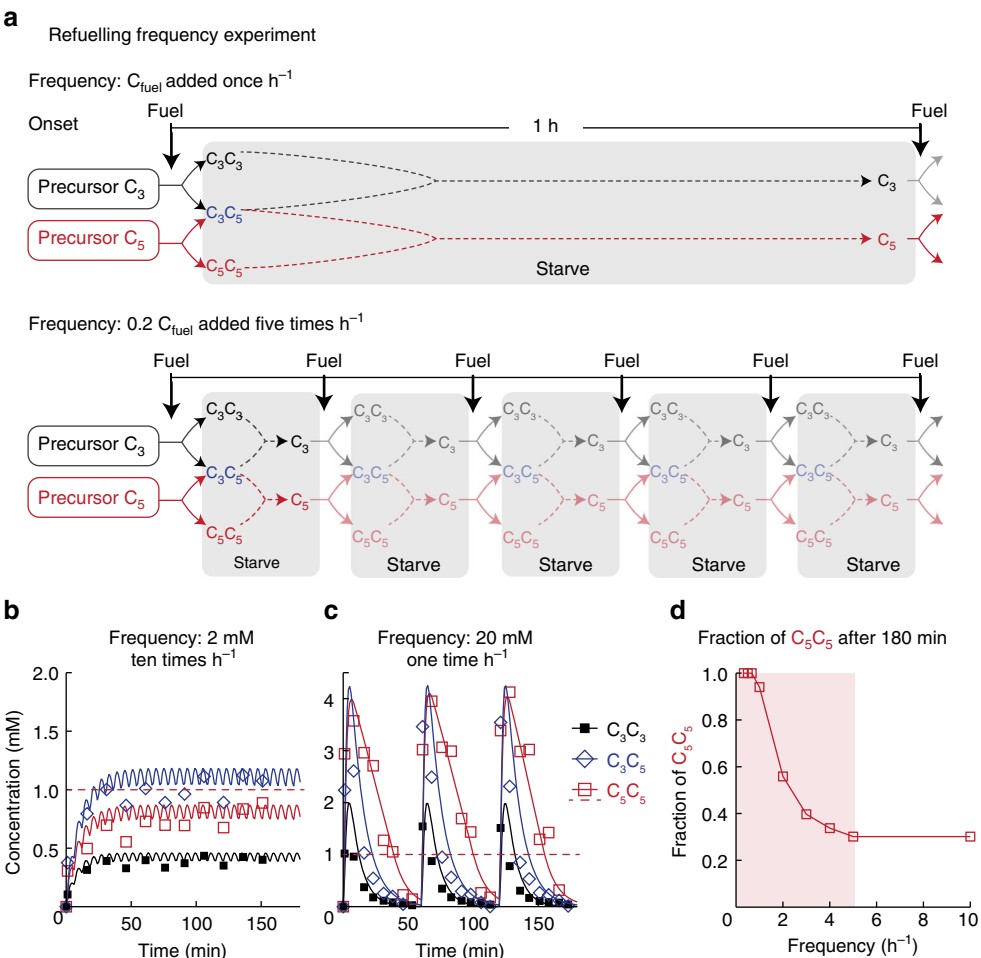

**Fig. 3** The method of fueling affects the selection process. **a** Minimalistic representation of the chemical reaction networks in competition experiments with varying frequency of fueling. Solutions of 300 mM $C_3$ and 300 mM $C_5$ are fueled with equal fuel flux, but varying frequency. Fuel can be added once per hour (top, 1 h$^{-1}$) or five times per hour in smaller batches (bottom, 5 h$^{-1}$). **b**, **c** The concentration of anhydride products against time when a mixture of 300 mM $C_3$ and 300 mM $C_5$ is fueled 60 mM EDC in a 3h experiment. The fuel was either supplied **b** with a frequency of 1 × 20 mM h$^{-1}$ or **c** 10 × 2 mM h$^{-1}$. Markers represent HPLC data; solid lines represent calculated data. The red dashed line indicates the solubility of $C_5C_5$. **d** Plot of the fraction of $C_5C_5$ after 3 h against the frequency of fuel addition. Only at a frequency below 5 h$^{-1}$ phase-separation occurred (gray-shaded area)

300 mM $C_3$ and 300 mM $C_5$. The competition experiments between $C_3$ and $C_6$ were carried out at 100 mM $C_3$ and 100 mM $C_6$. We performed the competition experiment between all precursors with 100 mM $C_3$, 100 mM $C_4$, 100 mM $C_5$, and 100 mM $C_6$. We carried out all experiments at 25 ± 0.5 °C.

**Continuous fueled kinetic experiments**. We used a microsyringe pump to supply a continuous flow of EDC (0.6 M) at 50 µL h$^{-1}$ to a 1.5 mL solution of 300 mM $C_3$ and 300 mM $C_5$. We stirred the mixture gently. We carried out all experiments at 25 ± 0.5 °C.

**HPLC**. We monitored the concentration profiles of the chemical reaction networks over time using analytical HPLC (HPLC, Thermo Fisher Dionex Ultimate 3000, Hypersil Gold 250 × 4.8 mm). We prepared a 1.5 mL sample as described in HPLC vial. The HPLC-injected samples of this solution, without further dilution, into the column, and all compounds involved were separated using linear gradients of water and acetonitrile. We used the following gradients. Method 1: water: acetonitrile gradient from 98:2 to 50:50 in 12 min. Method 2: water: acetonitrile gradient from 98:2 to 2:98 in 25 min. Method 3: water: acetonitrile gradient from 98:2 to 2:98 in 13 min. A flow of 2 min 98% acetonitrile followed all methods.

We prepared calibration curves for EDC, precursors, anhydrides, and *N*-acyl ureas in triplicate ($\lambda$ = 220 nm) to quantify the compounds over time. In the case of nonsymmetric anhydrides, which were not commercially available, absorbance values were assumed to be the average of their corresponding symmetric anhydrides, respectively. For instance, in the case of $C_3C_6$ anhydride, the calibration value from the average between the $C_3C_3$ and $C_6C_6$ anhydride values was used. See Supplementary Table 1-2 for all data.

**UV/Vis spectroscopy**. The UV/Vis measurements were carried out using a Multiskan FC (Thermo Fisher) microplate reader. Samples were directly prepared into 96-well plate (tissue culture plate non-treated) as described above. We performed all measurements in triplicate. We recorded the data at 600 and 400 nm at 25 ± 0.5 °C.

**Confocal fluorescence microscopy**. Confocal fluorescence microscopy was performed on a Leica SP5 confocal microscope using a ×63 oil immersion objective. We prepared samples as described above but with 2.5 µM Nile Red as a dye. A volume of 20 µL of the sample was deposited on the PEG-coated glass slide and covered with a 12 mm PEG-coated coverslip. A 543 nm laser excited the samples and we imaged at 580–700 nm. We replaced the solutions every 5 min. Every minute, we acquired a 4096 × 4096 image of an area that covered 246 × 246 µm. We performed each experiment in triplicate ($n$ = 3).

**Image analysis**. We used ImageJ's preinstalled "analyze particles" package, which is used to analyze the number of droplets and their circumference. The circumference was used to calculate their radii under the assumption that the drops were perfectly spherical. The acquired data were then binned in 5-min bins to ensure sufficient droplets per data point. Thus, each data point corresponds to five images (e.g., 1–5 min). We performed each experiment in triplicate ($n$ = 3).

**Kinetic model**. We wrote a kinetic model in Matlab that calculates the concentration of each reactant (carboxylic acid, *O*-isoacylurea, *N*-acyl urea, and anhydride) every second in a reaction cycle. The model calculated the concentrations of all relevant components on the basis of six differential equations that described six chemical reactions: direct hydrolysis of EDC ($r_0$), the activation of

acid ($r_1$), attack of the $O$-isoacylurea by a second acid to form the anhydride ($r_2$), hydrolysis of the $O$-isoacylurea ($r_3$), $N$ to $O$ shift of the $O$-isoacylurea ($r_4$) and the hydrolysis of the anhydride ($r_5$) (Supplementary Figure 1). We calculated the hydrolysis rate by the rate constant ($k_5$) multiplied by the concentration anhydride when the concentration anhydride was below its solubility. When the concentration anhydride was above its solubility, we calculated the hydrolysis rate by $k_5$ multiplied by its solubility ($s$).

We determined empirically or fitted the rate constant for all reactions. We determined the rate constant for the direct hydrolysis of carbodiimide ($k_0$) with a first-order rate constant of $1.3 \times 10^{-5}\,\text{s}^{-1}$ in earlier work by HPLC[18]. The rate constant for the formation of the $O$-acyl urea by reaction with EDC ($k_1$) was determined for each precursor by monitoring the EDC consumption with HPLC. This second order rate constant was independent of the nature of the precursor. The formation of the anhydride with a second order rate constant could not be determined ($k_2$) because the $O$-acyl urea was never observed. It was therefore set to be ten times the rate of $k_1$. As a result, the $O$-acyl urea never reached concentrations over 1 μM in the model. The rate constant for the direct hydrolysis of the $O$-acyl urea ($k_3$) could not be determined empirically because the $O$-acyl urea was not observed. The ratio of $k_2$ and $k_3$ was thus varied to fit the HPLC data for several concentrations of [EDC]$_0$ and [$C_n$]$_0$. The rate constant for the formation of the unreactive $N$-acyl urea ($k_4$) due to the rearrangement of the $O$-acyl urea was determined for each $N$-acyl urea by fitting the HPLC data. The first-order rate constant for the hydrolysis of anhydride ($k_5$) with a (pseudo)-first-order rate was determined by HPLC by determining the slope of the anhydride concentration in a regime where: (1) all EDC had been consumed and (2) no assemblies where present.

**Determining the anhydride solubilities**. The solubilities of the anhydrides could not be determined empirically due to their short half-life. Instead, we determined them by fitting the kinetic model to the HPLC data. First, $k_5$, the first-order rate constant for the anhydride hydrolysis, was determined by fitting the linear decay of $ln$[anhydride] against time in a regime where: (1) all EDC had been consumed and (2) no assemblies where present. The $k_5$ was determined for each anhydride for several runs. When $k_5$ was determined, the linear decay in the [anhydride] against time was fit in a regime where: (1) all EDC had been consumed and (2) assemblies where present. We fitted the slope by $r_5 = k_5 * s$. In which $s$ was the solubility of the anhydride.

**Calculating the persistence factor**. The kinetic model allowed us to calculate the anhydride hydrolysis rate with and without the inhibition mechanism in place. For each precursor, the maximum anhydride hydrolysis rate in response to 50 mM EDC was calculated, with or without the inhibition mechanism. These values allowed us to calculate by which degree the hydrolysis rate was decreased as a result of the inhibition mechanism. The maximum hydrolysis rate with inhibition mechanism was divided by the maximum hydrolysis rate without feedback to calculate the persistence factor.

For $C_3$, the factor was 1.0 because $C_3$ did not phase-separate and inhibit the deactivation. For $C_4$, the factor was 1.26, implying that in the presence of assemblies the hydrolysis was 1.26 times slower as compared to without assemblies. For $C_5$ and $C_6$, we found values of 18.4 and 118, respectively.

**MS**. We used a Varian 500 MS LC ion trap spectrometer to perform ESI–MS measurements. We diluted the samples in acetonitrile before injection into an acetonitrile carrier flow (20 μL min$^{-1}$).

**Data availability**. The data that support the findings of this study are available from the corresponding author upon reasonable request. Correspondence and requests for materials should be addressed to Prof. Job Boekhoven (job.boekhoven@tum.de).

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

## Acknowledgements

The authors are grateful for valuable discussions with Prof. Dieter Braun and Prof. Erwin Frey. This work was supported by the Technische Universität München—Institute for Advanced Study, funded by the German Excellence Initiative and the European Union Seventh Framework Programme under grant agreement no. 291763. B.R. and A.R.B

acknowledge funding by Deutsche Forschungsgemeinschaft within the SFB No. 863. M. T.S. acknowledges the European Union's Horizon 2020 Research and Innovation program for the Marie Sklodowska Curie Fellowship under grant agreement no. 747007. J.B. acknowledges funding by Deutsche Forschungsgemeinschaft the International Research Training Group ATUMS (IRTG 2022).

## Author contributions

J.B. and M.T.S. conceived the research. J.B., A.R.B. and M.T.-S. designed the experiments and wrote the manuscript. M.T.S., B.R., C.W. and J.B. performed experiments and analyzed the data.

## Additional information

**Competing interests:** The authors declare no competing interests.

