## [Peer Review File · Nature Communications]

Editorial Note: This manuscript has been previously reviewed at another journal that is not operating a transparent peer review scheme. This document only contains reviewer comments and rebuttal letters for versions considered at Nature Communications. Mentions of the other journal have been redacted.

REVIEWERS' COMMENTS:

Reviewer #1 (Remarks to the Author):

The authors have adequately modified the manuscript in response to the reviewers' comments on a prior submission to [redacted]. In my opinion the manuscript in its current form is suitable for publication in Nature Communications.

Reviewer #2 (Remarks to the Author):

I believe that the authors have addressed the technical comments that I made in my original review for [redacted]. I have no further comments or questions.

In my opinion, the manuscript in its present form is of suitable general interest for Nature Communications and I recommend its acceptance.

Reviewer #3 (Remarks to the Author):

The authors stressed the significance of preparing a system that expresses the characteristic properties of a living system, and analyzed it based on a bio-inspired concept. Undoubtedly, this is one of the most important fields of contemporary Chemistry. Although the revised manuscript was significantly improved, I would not recommend this manuscript for publication in [redaction].

The reasons are as follows.

1) A reaction system reported by Boekhoven and others is well designed and beautiful, but the target dynamics as a living system should be more specifically focused. As I commented previously, the gross result of this reaction system is not an unexpected one. It is no wonder that acid anhydride with a longer alkyl chain accumulates, associated with a phase change to an oil droplet. For example, chemists working in a supramolecular chemistry reported more advanced dynamics using micelles or vesicles. They picked up fatty acids with right lengths to have low CAC (critical aggregates concentration), and created even a vesicle which exhibited a self-reproductive dynamics and so forth.

2) The authors stress that importance of the chemical reaction network. I understand 4 reactants and 4 intermediates and many products are involved, but the ratio of various products is mainly determined by statistically, I presume. In a biological system, however, researchers are interested in more complicated chemical reaction networks. For example, reaction C is dependent on the product of reaction B, and reaction B is dependent on the product A, etc. I believe this kind of non-linear reaction should be investigated as a model study to reveal complex enzymatic reaction systems.

After all, I understand that a model reaction system proposed by Boekhoven and others are excellent and rationally analyzed as a material chemistry. In this sense, the manuscript is better to submit to a more specialized journal.

Response to Reviewers' comments:

Reviewer #1 (Remarks to the Author):

The authors have adequately modified the manuscript in response to the reviewers' comments on a prior submission to [redacted]. In my opinion the manuscript in its current form is suitable for publication in Nature Communications.

Boekhoven et al.: We thank the reviewer for his or her time.

Reviewer #2 (Remarks to the Author):

I believe that the authors have addressed the technical comments that I made in my original review for [redacted]. I have no further comments or questions.

In my opinion, the manuscript in its present form is of suitable general interest for Nature Communications and I recommend its acceptance.

- *Boekhoven et al.: We thank the reviewer for his or her time.*

Reviewer #3 (Remarks to the Author):

The authors stressed the significance of preparing a system that expresses the characteristic properties of a living system, and analyzed it based on a bio-inspired concept. Undoubtedly, this is one of the most important fields of contemporary Chemistry. Although the revised manuscript was significantly improved, I would not recommend this manuscript for publication in [redacted].

The reasons are as follows.

- *Boekhoven et al.: We thank the reviewer for his or her time.*

1) A reaction system reported by Boekhoven and others is well designed and beautiful, but the target dynamics as a living system should be more specifically focused. As I commented previously, the gross result of this reaction system is not an unexpected one. It is no wonder that acid anhydride with a longer alkyl chain accumulates, associated with a phase change to an oil droplet. For example, chemists working in a supramolecular chemistry reported more advanced dynamics using micelles or vesicles. They picked up fatty acids with right lengths to have low CAC (critical aggregates concentration), and created even a vesicle which exhibited a self-reproductive dynamics and so forth.

2) The authors stress that importance of the chemical reaction network. I understand 4 reactants and 4 intermediates and many products are involved, but the ratio of various products is mainly determined by statistically, I presume. In a biological system, however, researchers are interested in more complicated chemical reaction networks. For example, reaction C is dependent on the product of reaction B, and reaction B is dependent on the product A, etc. I believe this kind of non-linear reaction should be investigated as a model study to reveal complex enzymatic reaction systems.

After all, I understand that a model reaction system proposed by Boekhoven and others are excellent and rationally analyzed as a material chemistry. In this sense, the manuscript is better to submit to a more specialized journal.

- *Boekhoven et al.: We appreciate the advice and have transferred the manuscript to Nature Communications.*